# Current Technologies for Detection of COVID-19: Biosensors, Artificial Intelligence and Internet of Medical Things (IoMT): Review

**DOI:** 10.3390/s23010426

**Published:** 2022-12-30

**Authors:** Irkham Irkham, Abdullahi Umar Ibrahim, Chidi Wilson Nwekwo, Fadi Al-Turjman, Yeni Wahyuni Hartati

**Affiliations:** 1Department of Chemistry, Faculty of Mathematics and Natural Sciences, Padjadjaran University, Bandung 40173, Indonesia; 2Department of Biomedical Engineering, Near East University, Mersin 99138, Turkey; 3Research Center for AI and IoT, Faculty of Engineering, University of Kyrenia, Mersin 99138, Turkey; 4Artificial Intelligence Engineering Department, AI and Robotics Institute, Near East University, Mersin 99138, Turkey

**Keywords:** biosensors, COVID-19, artificial intelligence, computer-aided detection (CAD), Internet of Medical Things (IoMT)

## Abstract

Despite the fact that COVID-19 is no longer a global pandemic due to development and integration of different technologies for the diagnosis and treatment of the disease, technological advancement in the field of molecular biology, electronics, computer science, artificial intelligence, Internet of Things, nanotechnology, etc. has led to the development of molecular approaches and computer aided diagnosis for the detection of COVID-19. This study provides a holistic approach on COVID-19 detection based on (1) molecular diagnosis which includes RT-PCR, antigen–antibody, and CRISPR-based biosensors and (2) computer aided detection based on AI-driven models which include deep learning and transfer learning approach. The review also provide comparison between these two emerging technologies and open research issues for the development of smart-IoMT-enabled platforms for the detection of COVID-19.

## 1. Introduction

The year 2020 has witnessed a massive global burden due to the spread of the pneumonia causing virus known as SARS-CoV2 or COVID-19. The disease has led to massive screening, quarantines, restriction of movement, closure of land and borders, lockdowns, closure of educational, sportive and entertainment centers, and forced people to work from home [1,2]. In order to control the disease, scientists from different fields work hand in hand together to develop diagnostic approaches, prediction models, treatment control strategies, vaccines, etc. Screening of SARS-CoV-2 using a lab-bench assay is regarded as the first line of action in terms of minimizing spread and allowing for early treatment of the disease. This prompted the Chinese government to enact several testing points [2,3,4].

Medical experts rely on two main molecular approaches, which include RT-PCR and antibody–antigen based techniques, for the detection of the disease. However, among these two molecular testing approaches, RT-PCR is regarded as the gold standard technique due to it specificity and accuracy. The tests allow healthcare experts to detect viral nucleic acid from patient samples collected using a nasal swab, which is amplified using PCR machine. Antigen–antibody revolves around the binding between a synthesized recombinant antigen and the antibodies present in the body, which elicit an antigen–antibody reaction [5,6,7].

Even though molecular techniques such as RT-PCR and antibody testing are regarded as the standard procedures for the detection of SARS-CoV-2, they are hindered by several challenges, which include the incidence of false positive results, which can lead to misdiagnosis. These testing procedures are also costly, especially in remote areas and countries with substandard healthcare systems. As an alternative, healthcare professionals employ radiographic screening using X-ray imaging and CT-scan imaging, which allow scientists to discriminate between positive and negative cases. Others employ these techniques as a follow-up approach or as confirmation tests. As a result of massive or large-scale screening of radiographic images, these techniques can be tedious for radiologists and can led to misinterpretation [3,4,8,9].

In order to address these issues, scientists merge radiographic imaging with computer applications to develop computer-aided diagnosis (CAD), which allows screening of thousands of images with high accuracy, precision, and specificity [10,11]. CAD has been shown to aid medical experts in the past in the detection of different types of cancer, such as breast cancer [12], colon cancer [13], prostate cancer [14], brain cancer [15], tuberculosis [16], bacterial pneumonia [17], non-COVID-19 viral pneumonia [18], and skin diseases [19].

The integration of IoT with medical care, known as IoMT, is changing the landscape of patient care, diagnosis, and treatment. IoMT revolves around the interconnection between medical devices using internet. The platform enables machine–machine communication, patient–machine communication, machine–medical professional communication, etc. Examples of IoMT system include patient tracking devices, remote patient monitoring, medication tracking devices, etc. [20,21,22].

The prospect of smart diagnosis has been gaining ground in the last decade. The integration of smart technologies, such as AI and IoMT, with conventional diagnostic approaches has the potential to improve diagnosis and real-time, or point-of-care, detection, as well as to minimize errors and allow the sharing of medical data between devices, end users, and hospital cloud systems [23,24].

### 1.1. Comparison with Similar Studies

Detection of COVID-19 has been crucial for treatment and for controlling the spread of the virus. Scientists employ several emerging technologies, which include the use of CAD-based-on-AI driven models, AI/IoT enabled systems, molecular testing based on RT-PCR, and CRISPR/Cas based biosensors. Innumerable studies in the literature only concentrate on one or the other of these emerging technologies. However, this study evaluated each technique separately and compared them in terms of cost, performance (accuracy, sensitivity, specificity), deployment, etc.

The review provided by Samson et al. [25] focused on the application of biosensing technology for the detection of COVID-19. The review discusses nucleic acid-based biosensors, such as CRISPR/Cas9 strip-based biosensors, aptamer-based biosensors, surface plasmon resonance, and antigen–Au/Ag nanoparticles-based biosensors, as well as existing challenges and future perspectives. However, the review differs from the current study in terms of radiographic detection of COVID-19, AI-powered detection, and IoT-enable detection of COVID-19.

The study conducted by Santiago et al. [26] presented trends and innovations in biosensing technology for the detection of COVID-19. The study covers several molecular testing approaches, which include antigenic and serological rapid testing, as well as CRISPR-based biosensors. However, the study does not cover the use of medical imaging technology for the detection of COVID-19, computer aided detection, and IoT-enabled detection of COVID-19.

Another review that focusses on molecular and conventional testing was provided by Falzone et al. [27]. The study covers a wide range of diagnostic assays ranging from rapid antigen testing, antibody-based detection, immunoenzymatic serological testing, and RT-PCR. Other techniques covered include CRISPR/Cas-based approaches, nucleic acid amplification techniques, and digital PCR methods. The review also covers challenges and future perspectives.

The review conducted by Huang et al. [28] focuses on the application of AI-Powered detection of COVID-19 (which include ML, DL, and TL) using medical data such as electronic medical records and medical images (X-ray, CT scans, and ultrasound). The review also highlighted current challenges and future perspectives. Some of the topics not covered include molecular testing, biosensing technology, IoT-enabled detection, and comparisons between AI-powered systems and biosensors. The summary of the comparisons with similar studies is provided in Table 1.

### 1.2. Scope

The main aim of this review is to provide a holistic approach on emerging technologies that aid in the detection of COVID-19, such as RT-PCR, antigen–antibody, and CRISPR-based biosensors, as well as computer aided detection using AI-driven models. Moreover, the review also covers the integration of IoMT and AI for the development of a smart system for the detection of the disease.

Section 2 discusses the COVID-19 pandemic. Section 3 presents an overview on molecular approaches for the detection of COVID-19 using RT-PCR and CRISPR-based biosensors. Section 4 discusses computer aided detection of COVID-19 from radiographic images. Section 5 presents diagnostic imaging, which includes X-rays and CT scans. Section 6 discusses comparisons between molecular approaches and computer aided detection, as well as smart AI/IoMT-enabled platforms for the detection of the disease. Section 7 discusses open research issues and concluding remarks.

## 2. COVID-19

For the first time in a century, the world witnessed another global pandemic caused by a coronavirus known as Severe Acute Respiratory Syndrome Coronavirus (SARS-CoV-2). The virus is traced back to a sea food market in Wuhan, Hubei province, China, in late December, 2019. Coronaviruses are positive, single-stranded RNA viruses which belong to the Coronaviridae family, which also include SARS-CoV-1 and MERS-CoV. Both the two coronaviruses have caused epidemics and endemics in the last few years [1,29].

SARS-CoV-1 was first identified in the year 2003 in China, where bats are regarded as the main reservoirs. The virus has spread to four other countries, causing global epidemics. SARS-CoV-1 affected close to 8000 people with an approximately 10% mortality rate. The World Health Organization, along with other international and non-Governmental organizations, collaborated to control and prevent further spread of the virus [30,31].

Middle East Respiratory Syndrome Coronavirus (MERS-CoV) is another virus that belongs to Coronaviridae family that caused a global burden in the year 2012. The virus was first identified in Saudi Arabia, and later spread to 27 countries, leading to approximately 2600 cases. The disease has infected thousands, and approximately 35% of patients died from the disease. Dromedary camels were linked with the transmission of the virus to human primates [32,33]. Figure 1 shows the differences and similarities between SARS-CoV-1, MERS-CoV-2, and SERS-CoV-2 in terms of number of cases, fatality rate, mortality rate, reservoir, etc.

### 2.1. Transmission of SARS-CoV-2

Unlike previous coronavirus diseases that are associated with animal transmission, such as bat in SARS-CoV-1 and dromedary camels in MERS-CoV, no animal reservoir has been found for SARS-CoV-2 [34]. Several studies have shown that the virus can be spread directly from one person to another (Human–human transmission) via sneezing or coughing, or indirectly, such as by coming in contact with surfaces infected with the virus [34,35].

### 2.2. Symptoms of SARS-CoV-2

The clinical spectrum of the COVID-19 disease ranges from asymptomatic to severe acute respiratory disease and death. People infected with the disease display pneumonia symptoms, which include shortness of breath, sore throat, fever, fatigue, and cough. People that are at risk of COVID-19 include elderly people who are suffering from chronic diseases, such as chronic lung disease, cancer, hypertension, renal and kidney diseases, diabetes, cardiovascular diseases, etc. [35,36]. The clinical manifestations of SARS-CoV-2 are presented in Figure 2 (redesigned using Biorender [27]).

## 3. Molecular Diagnosis of COVID-19 

Early and accurate detection of COVID-19 disease is crucial for timely management and prevention. The field of disease detection has been transformed from conventional diagnosis, such as microscopy, with lower sensitivity and specificity, to molecular diagnosis, such as antigen–antibody, enzyme-substrate and NA probe-target, and biosensing technologies [5,37], as shown in Figure 3. Diagnostic imaging based on CT scans, X-rays and ultrasound imaging are currently in use as an alternative or confirmatory approach for the detection of COVID-19 [38,39].

### 3.1. Laboratory Assays

Accurate, sensitive, and rapid laboratory assays for the detection of SARS-CoV-2 are crucial for the treatment and control of COVID-19 infection. Currently, there are a myriad of tests available in the market. However, the adoption of appropriate laboratory testing techniques and types of specimen (nasopharyngeal aspirates, nasopharyngeal swabs, mid-turbinate swabs, oropharyngeal swab, etc.) is one of the cornerstones for the timely management and control of the disease. The literature encompasses several studies focusing on laboratory testing procedures, which include nucleic acid amplifications, antigen tests, antibody tests, and point-of-care testing. These procedures are currently employed in the detection of SARS-CoV-2 in clinical diagnosis of symptomatic patients, asymptomatic population screening, contact investigations, targeted high-risk population screening, retrospective population screening, monitoring of infectivity, disease severity monitoring, etc. [38,39,40].

#### 3.1.1. Reverse Transcription-Polymerase Chain Reaction (RT-PCR)

RT-PCR is regarded as the most reliable approach for the detection SARS-CoV-2 [5,41,42]. RT-PCR is a nuclear-derived approach for the detection of the genetic content of pathogens, such as viruses (such as Zika and Ebola) and bacteria. The early RT-PCR testing employ radioisotope markers which are subsequently replaced by fluorescent dyes. As shown in Figure 4, the procedure for conducting RT-PCR test follows four steps:

Sample collection: Nasal swab and nasopharyngeal samples are collected by medical experts. The samples are sealed and transported to the laboratory for detection.

Extraction: This step allows medical technologies to extract or isolate viral NA. This stage revolves around the use of chemicals to remove components such as fats and proteins.

PCR: The isolated viral NA is further amplified using a PCR machine, also known as a thermal cycler. PCR machines amplify thousands of complies of the viral NA, which increases the sensitivity and specificity of detection. Reverse transcription is carried out in order to convert RNA strands of the virus to DNA.

Detection: After the RNA is transcribed to DNA and amplified, the machine detects the presence of virus DNA due to the release of fluorescent dye, which can be measured in real-time and presented on the computer screen.

The RT-PCR method is highly specific and sensitive compared to the antigen–antibody method. The test can take between 3 and 6 h to process and obtain results. Moreover, this approach has shown to be faster, reliable, and to present a lower rate of errors or false positive results compared to other approaches. One of the disadvantages of RT-PCR is that it can’t be used to identify past diseases, which is crucial for understanding the pathology and spread of the diseases [43,44].

#### 3.1.2. Antibody-Based Method

The emergence of SARS-CoV-2 has prompted scientists to develop viable diagnostic assays that can accurately detect the presence of the virus from biologically derived samples. The antigen–antibody approach, as the second standard approach, revolves around the binding between a synthesized recombinant antigen (produced in the laboratory which mimics specific structures of SARS-CoV-2) and the antibodies present in the body, which elicit an antigen–antibody reaction. Unlike the RT-PCR, the specificity of antigen–antibody testing approach relies on the affinity of target antigen designed. Therefore, designing an antigen specific to the antibodies produced as a result of the present of the virus is crucial for increasing specificity and minimizing the probability of false positive results [45,46].

#### 3.1.3. Antigen-Based Method

The COVID-19 antigen test is another popular diagnosis approach, used as an alternative to the RT-PCR approach. This test is mostly used for early detection of the disease and determining if a patient is contagious. COVID-19 antigen testing revolves around the use of different type samples, which include oral, nasal, and respiratory tracts. The advantages of this approach are that it is very easy to operate, and that it can be used for early detection (i.e., 2 days before the onset of the symptoms). Antigen-based method can be divided into lateral flow rapid-test cassette format and enzyme-linked immunosorbent assay (ELISA) [46,47].

The lateral flow assay-based COVID-19 antigen test, also known as the antigen rapid test, revolves around the collection of plasma, serum, or blood bleed from the fingertip of suspected person, which is subsequently transferred to the test cassette. The test lasts for 20 min, after which the result is displayed. Compared to the lateral flow assay, ELISA-based COVID-19 antigen tests produce accurate and more reliable results. One of the disadvantages of this approach is the requirement of an intensive laboratory procedure, which can be challenging for remote areas with limited healthcare resources [46,48].

### 3.2. Strengths and Weakness of Molecular Testing

Currently, there are several techniques developed for the detection of pathogens (such as viruses and bacteria). Among these techniques, molecular testing is the most prepared approach, due to its high sensitivity and specificity. These molecular tests include antibody, antigen, and RT-PCR, which can also be subdivided into the molecular test and the serological or antibody test. Molecular testing revolves around the detection of viral RNA in the human body while the virus is still replicating, while antibody assays detect the presence of antibodies produced as a result of human immune response against the virus [7]. In terms of specificity, the antibody assay has a higher specificity, as it can detect if a patient has had COVID-19. One of the limitations of this approach is that antibodies may not be detectable until 1–3 weeks after infection. The antigen test is one of the most rapid and simple technique that can be used to detect SARS-CoV-2 in both asymptomatic and symptomatic patients. Just like the antibody test, the virus can be detected between 5 and 12 days after contact, or after onset of symptoms, and results can be generated after 15 min [49,50]. Despite the test being rapid, fast, and simple, it is hindered by several challenges, which include low sensitivity compared to RT-PCR and the likelihood of false negative results [51].

RT-PCR is the most preferred approach, as it can detect the presence of the virus in an early stage of infection. Even though this test is regarded as the standard approach, one of its limitations is the likelihood of false negative results. Thus, repeated testing is required to overcome this challenge, which increases testing time and cost. Quantitative RT-PCR (RT-qPCR) is another highly sensitive molecular method that can be employed for the detection of COVID-19. The method revolves around the combination of real-time quantification and RT-PCR using fluorescent probes. However, this technique is less widely use due to its expensive instrumentation [52,53].

### 3.3. Application of Biosensors for the Detection of SARS-CoV-2

The application of a rapid, ultra-sensitive, and quantitative electrochemical biosensor for the detection of SARS-CoV-2 was proposed by Alateef et al. [54]. The biosensor is designed using gold NPs (AUNPs) which is capped with highly specific antisense oligonucleotide, while the sensing probe was immobilized on a paper-based electrochemical device. The sensing mechanism revolves around the interaction between the antisense SSDNA and sensing probe, which generate readout results that can be seen on a hand-held reader. In order to analyze the sensing viability of the paper-based electrochemical biosensor, the developed platform was tested using clinical and vero cells infected with the virus. The performance evaluation of the biosensor resulted in sensitivity of 231 (copies µL^−1^)^−1^ and 6.9 copies/UL LOD. Subsequent testing of the device using both samples obtained from 22 patients tested with SARS-CoV-2 confirmed using RT-PCR and 26 healthy patients resulted in 100 accuracy, sensitivity and specificity.

The development of a cheap CRISPR-based POC testing platform known as miSHERLOCK for the detection of SARS-CoV-2 was proposed by de Puig et al. [55]. The sensing mechanism revolves around the collection of unprocessed saliva, followed by extraction, purification, concentration, amplification, and detection based on the interaction between viral NA and guide RNA binds with Cas12a to produce fluorescent visual output within 1 h. The performance of the platform resulted in highly sensitive and multiplexed detection of the virus and mutations associated with two different variants, leading to different LOD in cp/mL. Another distinction of this approach is the application of adjunct smartphones to enable quantification of output, automated interpretation, and the prospect of remote and distributed result reporting.

Song et al. [56] developed an antifouling electrochemical biosensor for the detection of SARS-CoV-2 NA. The nanobiosensor is designed based on electropolymerized polyaniline nanowires and a synthesized Y-shaped peptide which poses antifouling properties. The mechanism behind the working principle of the biosensor revolves around the interaction between immobilized biotin-labeled probes and COVID-19 NA. Evaluation of the performance of the genosensor led to a 3.5 fM detection limit and a wide linear range of 10–14 to 10–9 M.

Detection of SARS-CoV-2 from clinical samples using a field-effect transistor (FET)-biosensor was proposed by Seo et al. [57]. The biosensor was constructed by coating a graphene sheet of the FET with a specific antibody against the viral spike protein. In order to test the viability of the immunobiosensor, several samples, such as antigen proteins, cultured virus, and nasopharyngeal swab samples collected from patients suffering from COVID-19 pneumonia. The performance of the FET-based biosensor for the detection of SARS-CoV-2 spike protein yielded 100 fg/mL concentration in clinical transport medium and 1 fg/mL concentration in phosphate buffer saline. The biosensor was able to detect SARS-CoV-2 in cultured medium with 1.6 × 101 pfu/mL LOD and clinical samples with 2.42 × 102 copies/ML.

Tian et al. [58] developed an electrochemical aptamer-based biosensor for the detection of COVID-19. The biosensor was constructed using metal-organic frameworks MIL-53(AI) which is decorated using AU@Pt NPs and enzymes. The surface of the electrodes was immobilized with dual aptamer as biorecognition element. SARS-CoV-2 is detected based on the interaction between immobilized 2 thiol-modified aptamers (N48 and N61) and SARS-CoV-2 nucleocapsid via the co-catalysis of the nanomaterials, G-quadruplex DNAzyme, and Horseradish Peroxidase (HRP). Evaluation of the biosensor demonstrated 8.33 pg mL^−1^ LOD and a wide linear range of 0.025 to 50 ng ML^−1^.

Buyuksunetci et al. [59] developed an electrochemical biosensor for the detection of SARS-CoV-2. The device was constructed using a gold screen printed electrode (AuSPE) and subsequently immobilized with either angiotensin-converting enzyme 2 or CD147. The biosensor is designed based on the interaction between the spike protein and receptors such (ACE2) or CD147. Evaluation of the performance of the biosensor yielded 29,930 ng ML^−1^ LOD and a linear detection range of 700 ng ML^−1^ to 1500 ng ML^−1^, and 1500 ng ML^−1^ to 7000 ng ML^−1^ for the detection of spike protein using ACE2, while detection of spike protein using CD147 yielded 38.99 ng ML^−1^ LOD and linear detection ranges of 500 ng ML^−1^ to 5000 ng ML^−1^. The biosensor was also evaluated using clinical samples confirmed with RT-PCR method.

The development of a multiplexed grating-couple fluorescent plasmonic biosensor for the detection of SARS-CoV-2 using either a dried blood spot sample or human blood serum was proposed by Cady et al. [60]. Detection of COVID-19 relied upon the interaction between antibody (IgG) and antigen (nucleocapsid protein, Spike S1 and Spike S1 S2). The performance evaluation of the immunobiosensor produced linear response for serum samples diluted to 1:1600 dilution. The biosensor was also compared with two commercial COVID antibody testing kits (which include Luminex-based microsphere immunoassay and ELISA), which resulted in 100% correlation. Moreover, 63 samples of dried blood spots were tested using the constructed immunobiosensor, which yielded 86.7% sensitivity, and 100% selectivity for detection prior to COVID-19 infection.

Kim et al. [61] developed a sensitive electrochemical biosensor for point-of-care detection of COVID-19. The genobiosensor was designed using a multi-microelectrode array and relied upon the interaction between probes and target genes (N gene and RdRP gene) amplified using Recombinase Polymerase Amplification (RPA) and subsequently detected using pulse voltammetry. This process involved hybridization between thiol-modified primers immobilized on the surface of WE and RPA amplicon, which resulted in reduction of current density due to accumulation of amplicons. The performance of the assay yielded 3925 fg/µL LOD for N gene and 0.972 fg/µL for RdRP gene.

## 4. Computer-Aided Diagnosis (CAD) and Internet of Medical Things (IoMT)

Computer aided diagnosis (CAD) is regarded as one of the technologies that is transforming medical diagnostics. This technology revolves around the use of computer applications, software, and algorithms for detection of diseases that often require human expertise, prolong procedures, the use of chemicals or radiations, etc. [10]. CAD technology is driven by Machine Learning (ML), Deep Learning (DL) as sub-field of ML, and transfer learning, which allows the transfer of knowledge learned from trained networks to perform similar functions on different tasks [62].

The field of medical diagnosis is undergoing revolution due to the integration of CAD, automated detection, and smart sensing. Medical imaging based on diagnostic radiology is one of the major fields that is transforming to a more accurate, reliable, fast, cost-effective diagnostic. CAD is currently aiding medical experts in appropriate decision making. The history of application of CAD technology can be traced back to 1960s. However, it wasn’t until the 1980s when this technology started gaining ground due to the fundamental change in the approach on the use of computer output from automated computer diagnosis to CAD [10,11].

### 4.1. Artificial Intelligence (AI) and Machine Learning (ML)

The concept of AI is dated back to 19th century, when it started as a theory. The field has now exploded into different disciplines ranging from marketing, business and finance, advertisement, and smart devices, to agriculture and medical care. The transformation in the field of data storage and data analytics has driven the field and transformed our daily lives [63,64]. AI and ML are used interchangeably, and there are several misconceptions about the exact meaning of each concept. AI intelligence revolves around the use of a computer to mimic human cognitive functions, such as learning and problem solving or decision making. In other words, AI is the application of computer program that enables machine to perform specific tasks [63].

The concept of ML revolves around the use of algorithms whose performance improve as a result of exposure to large amounts of data over time. In ML, a series of algorithms is applied to allow computer to learn, analyze data, and make decisions based on the learned knowledge. ML models are either used for classification, regression, or clustering. An example of traditional ML models uses for classification include SVM and Naive Bayes classifier. Clustering ML models include K-means and tree-based clustering, while Linear regression, Random Forest, and KNN models are used for regression tasks [65,66]. ML models require large amounts of data in other to make appropriate decisions. Just like the way cars are driven by fuels, ML models are driven by large amounts of data. Some of the applications of ML can be found in Information Technology (IT) applications, weather forecasting, gaming, robotics, stockbroking, etc. [64].

ML models learn through a process known as gradient descent or loss function, where models minimize errors between a predictive value and the actual or ground truth value. After every iteration, the models compare the actual value with the objective or predictive value and adjust parameters so that the error becomes smaller [67]. ML algorithms can be classified into Supervised Machine Learning (SML), Unsupervised Machine Learning (UML), and Reinforcement Machine Learning (RML) [68].

#### 4.1.1. Supervised Machine Learning

Supervised Machine Learning (SML) is a branch under ML where computer algorithms are trained using labelled data. The ML models are trained using backpropagation until they can detect underlying patterns and the relationship between the input data and the labelled output. The model is subsequently evaluated using test sets (unseen or untrained datasets) [69]. SML has shown to achieve high performance, however, one of the challenges associated with this type of ML is “overfitting”, where models perform very well on the training set but perform poorly on test sets. Thus, scientists proposed several ways to counter this issue, through cross validation, data augmentation, regularization, the use of ensemble models, etc. [67,70].

There are several applications of SML which include classification and regression tasks. Several studies have reported the application of classification algorithms for detection of clinical diseases [65,66]. SML models can be used to classify diseases into binary cases (disease/healthy, positive/negative, findings/no findings etc.), ternary, and quaternary classification in the case of different grades of tumors etc. [71]. Regression models, on the other hand, produce numerical correlations between the input data and output data [66]. Several prediction models are used in healthcare settings for the prediction of disease and drug discovery [71].

#### 4.1.2. Unsupervised Machine Learning

Unsupervised Machine Learning (USML) is a sub-branch of ML where algorithms are trained using an unclassified or unlabeled dataset. Unlike SML, where data are labelled and models optimized between the predicted value and the actual value, USML learns or recognizes patterns in data and groups them or clusters them together. USML algorithms are trained using unsorted data, where models sort out the data based on similarities and differences. Another difference, and limitation, of USML is that they can be unpredictable compared to SML. Some of the advantages of USML include being less costly, faster, easier (which are associated with less manual work in labelling data), the ability to use real-time data, etc. [68,72].

Clustering algorithms are the most common USML algorithms used for clustering unstructured and unsorted data into different groups. Some of the classifications of clustering algorithms include hierarchal, overlapping exclusive, and probabilistic algorithms. Common examples of clustering algorithms include K-means clustering, Gaussian Mixture models, and Principal Component Analysis (PCA) [66,73]. USML are currently applied in healthcare settings for classification, segmentation and medical image detection, detection of anomalies in medical data, health index monitoring, drug discovery, genomics, etc. [65,74].

### 4.2. Deep Learning (DL)

Deep Learning (DL) is a subfield of ML which is inspired by how human brains function due to connections or synopsis of nerve cells or neurons. DL is a sub-field of ML which revolves around the use of multiple perceptron, where each layer of the network is connected to another layer. Just like ML, DL models learn from a vast amount of data, which is crucial for high performance in terms of accuracy [75,76].

One of the advantages of DL over ML models (known as flat models) such as decision trees, logistic regression, SVM, etc. is that DL models can take raw input in the form of images and texts without the need of preprocessing steps. Example of DL applications include Google translation, chatbots, self-driving cars, Netflix movie suggestions, and personal Assistant such as Siri and Alexa [75,77]. The current boom of big data is transforming DL models, which are powered by massive amounts of data. Another advantage of DL models over traditional ML models is that DL models tend to result in higher accuracy with an increase in the amount of training and testing datasets, while ML can become saturated and stop improving [76].

### 4.3. Internet of Medical Things (IoMT)

IoMT, also known as IoT of healthcare, revolves around the internet-connection of medical appliances, hardware, and software. The system enables the wireless connection between devices and servers, as well as storage of medical data in the cloud and subsequent analysis using AI-powered models. The application of IoMT is growing exponentially due to advancements in hardware and software engineering [20,21,22].

The applications of IoMT include in-hospital, in-home, and on-body. In-hospital IoMT revolves around the use of sensors to track patients and the transmission of medical data from one department to another or between hospital devices and physicians [78]. In-home IoMT revolves around the transfer of medical data between users and primary care providers stationed in healthcare settings. An example of an in-home IoMT is remote patient monitoring, where medical devices transmit medical data such as heart rate, blood pressure, and blood oxygen saturation to physicians for evaluation and decision making. On-body IoMT revolves around the use of wearable devices and implantable-IoT enabled devices connected with remote tracking systems or monitoring systems. Despite the wide application and potential of IoMT, it is limited by several challenges [79]. Some of these challenges include privacy concern, safety, and security [21,22,78].

#### 4.3.1. Advantage of IoT-Based Systems: How IoT Is Shaping Clinical Diagnosis

Recent advances in sensing technology, IT, and software engineering, and its adoption in healthcare settings, have made remote monitoring, real-time diagnosis, analysis, and sharing of data possible. This technology has shown to contribute in making diagnosis more efficient and safer, as well as aiding medical experts in making appropriate diagnoses. IoT/AI-driven models are applied in medical settings at massive scales in order to relieve the intensive workload of medical experts, to increase performance and efficiency, and to minimize long diagnostic processes [80]. Despite the advances made in the last few years, the world continues to record high mortality rates due to the lack of adequate and advanced healthcare facilities, such as conducive and hygienic environments, medical diagnostic kits, and medical devices, as well as the high cost of molecular and imaging diagnosis and the high rate of misdiagnosis. The application of AI-based systems (also known as CAD) in part helps address these issues. However, the integration of IoT-based systems with AI-driven models contributes to real-time diagnosis without the need of an experts or in-clinic diagnosis procedures [80,81].

#### 4.3.2. Disadvantage of IoT-Based Systems

Despite the fact that IoT offers several benefits in healthcare settings, one of the major challenges limiting its application are security threats, such as data theft, device hijacking, system attacks (e.g., Distributed Denial of Service or DDoS), data ownership disputes, etc. The interconnection between IoT-based devices with the internet makes them prone to cyber-attacks or hacks [82,83]. Moreover, the use of IoT-based devices requires the use of data storage, such as the cloud, as well as the transfer of a patient’s data through the internet, which raises issues regarding privacy and security [84]. In the last few years, several companies have developed encrypted approaches to prevent fraudulent attacks and breaches of privacy. Despite their efforts, IoT medical devices continue to be targets of cyber-attacks. To increase security levels, these companies developed an authentication approach which gives access to only a few staff members with clearance. Other practices that can increase security include implementing security protocols through tracking and monitoring, network segmentation, monitoring of inventories of devices, and encryption [82,83,84].

#### 4.3.3. Deployment of IoT-Based Systems

Deployment of IoT systems for diagnosis of pathological diseases requires several features, which include medical data (such as images acquired from CT scans, X-rays, MRIs, PETs, SPECTs, hybrid systems, electrophysiological devices, human faces, skin, etc.), AI-driven models trained and validated using large amounts of data, and websites that can be used to deploy the model for real-time classifications [80]. Currently, there are several IoT/AI-powered websites that allow users to upload pictures of their faces or skin taken using mobile phones for dermatological or disease detection. Handfuls of these platforms have been developed for the detection of COVID-19 using CT scan and X-ray images. However, these systems still require the use of radiographic images from clinical settings. Thus, IoT-based artificial intelligence systems have shown to serve as a confirmatory system that can be used to aid clinicians in making appropriate decisions [81,85].

The cost of deployment of IOT/AI-based systems depends on the type of infectious disease, the complexity of the system, and the requirement of medical devices for the generation of medical data. Image of skin and faces for dermatological-AI/IoT systems only require the use of mobile phones to capture images, and subsequent upload of the images into the system results in classification in real-time. While other systems rely on medical images generated from medical devices such as microscopes, X-rays, ultrasound, etc., which increases processing time and cost [86,87].

## 5. Diagnostic Imaging

Diagnostic imaging is a sub-field under medical diagnostic that allow medical technologies and radiologists to view the interior of the human body and to analyze the presence of injury or other health complications. This type of diagnostic uses several types of machines, which allow the reconstruction of structures inside the body. Some of these devices include MRIs, CT scans, X-rays, ultrasound, mammography, arthrogram, and bone density scan [79].

### 5.1. Radiographic Imaging of COVID-19

The field of medical imaging has transformed from conventional imaging such as X-rays, CT scans, MRIs, and ultrasound imaging, to nuclear imaging based on PET, SPECT, and hybrid imaging such as PET/CT, PET/MRI, SPECT/PET, SPECT/MRI etc. [10,79]. Medical imaging revolves around the application of different imaging modalities to help physicians diagnose several conditions affecting patients. The use of medical imaging devices allows medical experts to view internal organs and tissues, and confer diagnostics such as fracture, dislocation, cancer, pneumonia, tuberculosis, etc. [88]. Detection of pneumonia using X-ray images and CT scan machines has become an alternative or confirmatory test for detection of non-COVID-19 (such as bacterial and influenza viral) pneumonia and COVID-19 [4,88,89]

#### 5.1.1. X-ray Imaging

X-ray imaging is one of the most common techniques used in clinical and other healthcare settings for the diagnosis of a wide range of diseases. The advantages of X-ray imaging over other imaging techniques include low radiation, availability (i.e., due to high demand), low cost, moderate sensitivity, and low radiation dose [90]. The classification of chest X-ray images by radiologists include posteroanterior, anteroposterior, and lateral views, as shown in Figure 5. These classifications are based on the position and orientation of patient parallel to the X-ray source and detector panel. Side view or lateral view differs from both anteroposterior and posteroanterior (which are known as frontal views). The side view is obtained as a result of the combination of posteroanterior view and projection of the x-ray from one side of the patient to the other or right to left. The frontal views are based on the positioning of the X-ray source to the front or rear of the patient. Posteroanterior X-ray imaging is generated in erect standing position of the patients, while anteroposterior X-ray image is obtained from patients in the supine position [90,91].

#### 5.1.2. CT Scan Imaging

CT scan imaging modality is perceived by many scientists as the most efficient technique for screening for pulmonary diseases. One of the major differences between the CT scan machine and the X-ray machine is that X-ray machine uses a very small amount of radiation. The CT scan is more detailed, as it provides 3D images of tissues and organs, while X-ray provides 2D images. CT scan has shown to be more effective and sensitive in terms of imaging the chest, with outstanding spatial resolution. However, the limiting factors of using CT scan machine include exposure to high radiation and high cost [92]. The images of COVID-19 and normal cases are presented in Figure 6.

### 5.2. Radiographic Dataset

Ever since the WHO declared COVID-19 as global pandemic, medical experts have curated several radiographic datasets from clinical settings into an online repository. Kaggle and GitHub are among two of the most popular domains that are easily accessible. These repositories contain thousands of radiographic images of both X-ray and CT scan images of bacterial pneumonia, COVID-19 pneumonia, non-COVID-19 viral pneumonia, and healthy cases. One of the challenges of using collections of more than one dataset is the likelihood of reputation and the diversity of images acquired from different types of devices and settings.

(A)JP Cohen COVID-19 Xray Dataset

The dataset provided by Cohen et al. [93] is the most popular dataset used by scientists for CAD of COVID-19 and healthy cases. The dataset titled “covid-chestxray-dataset” comprises over 200 COVID-19 X-ray images, which are updated frequently on a GitHub repository. The images are curated from several medical and scientific platforms. The dataset contains both X-ray and CT scan images of COVID-19, MERS, SARS, ARDS, and other diseases. The images curated comprise four different views, which include lateral, anteroposterior, anteroposterior supine, and posteroanterior. The images are acquired from both male and female patients between the age range of 50 and 80 years old, and is available on https://github.com/ieee8023/covid-chestxray-dataset (accessed on 25 November 2022).

(B)COVID-19 Radiography Dataset

The COVID-19 radiography dataset is made available on the Kaggle repository (https://www.kaggle.com/tawsifurrahman/covid19-radiography-database, accessed on 25 November 2022). The dataset contains different collections of pneumonia cases, which include 3616 COVID-19 images, 1345 viral pneumonia X-ray images, 10,192 normal X-ray images, and 6012 lung opacity (non-COVID lung diseases). The dataset is curated by a group of scientists from different universities and medical institutions within Asia. The database is updated regularly. The first release contains 219 COVID-19, 1341 normal, and 1345 viral pneumonia chest X-ray (CXR) images. Since the first release, the database has been updated twice. In the first update, the COVID-19 CXR images were increased to 1200 images, and to 3616 in the second update.

(C)COVIDx Dataset

COVIDx dataset is provided by Wang et al. [94]. The dataset is made of a collection of two public dataset, which include an RSNA challenge dataset and a COVID-19 Image Data Collection. Unlike the one provided by [93] which contains binary classes, this dataset contains three classes, which include normal, non-COVID-19 viral, and COVID-19 pneumonia. The dataset comprises 13,800 total images acquired from 13 thousand people. The overall dataset is partitioned into training (13,569) and testing (231). The images are available at https://github.com/lindawangg/COVID-Net (accessed on 25 November 2022).

(D)HCV-UFPR COVID-19 Dataset

This dataset is made available by Hospital da Cruz Vermelha, in the southern part of Brazil. The dataset is composed of 281 COVID-19 and 232 normal CXR images. Unlike the rest of the dataset mentioned, the HCV-UFPR COVID-19 dataset is private, but access can be granted when requested [95].

(E)SARS-CoV-2 CT Scan Dataset

This dataset is provided by Soares et al. [96]. It is considered one of the largest CT scan datasets for both COVID-19 and non-COVID-19 CT scan images, with a total of 2481 (1252 COVID-19 and 1229 normal) images. The images are acquired from several patients in Sao Paulo, Brazil. The dataset is made available on both Kaggle and GitHub at www.kaggle.com/plameneduardo/sarscov2-ctscan-dataset (accessed on 25 November 2022) and https://github.com/Plamen-Eduardo/xDNN-SARS-CoV-2-CT-Scan (accessed on 25 November 2022), respectively.

(F)Chest X-ray

Prior to the COVID-19 pandemic, the dataset made available by Kermany et al. [97], which is available on Kaggle (https://www.kaggle.com/paultimothymooney/chest-xray-pneumonia, accessed on 25 November 2022), is the most widely used dataset for the classification of non-COVID-19 pneumonia cases. The dataset contains a sum of 5856 images, which are grouped into training, testing, and validation. The description of the dataset is curated according to X-ray images collected from retrospective pediatric patients between the ages of 1 and 5 years old.

(G)ChestX-ray8

The ChestX-ray8 is a large dataset curated by Wang et al. [98]. The dataset contains 108,948 frontal view X-ray images of 32,717 unique patients, classified into eight diseases, which include pneumonia, pneumothorax, effusion, mass, nodule, infiltration, atelectasis, and cardiomegaly. The database contains 24,636 X-ray images with one or more clinical diseases, while the remaining 84,312 X-ray images are healthy cases. The data is available at https://www.kaggle.com/nih-chest-xrays/data/home (accessed on 25 November 2022). Table 1 presents the summary of public accessible datasets.

### 5.3. AI-Powered Detection of COVID-19 from Radiographic Imaging

Since the first declaration of COVID-19 as a global pandemic by the world health organization, scientists all over the world have contributed immensely to the detection and prediction of the disease using AI-driven models. Computer aided diagnosis of COVID-19 is limited to the use of X-ray and CT scan images of patients suspected to have the diseases. Several AI-driven models have been deployed, which include the use of models developed from scratch, pretrained models, hybrid models, ensembled models, etc. [18].

The literature is copious, with several studies on the use of AI-driven models for the classification of COVID-19. Some of these studies conducted binary (two-way), ternary (three-way), and quaternary classifications. Considering the fact that there are several studies that conducted binary, ternary, and quaternary classifications in one article, this study will attempt to categorize these studies based on the highest number of classifications. 

#### 5.3.1. AI-Powered Detection of COVID-19 from X-ray Images

(A)Binary

The study conducted by Gayathri et al. [99] applied several pretrained networks and their combinations for detection of COVID-19 from X-ray images. The detection process revolves around the use of pretrained networks for feature extraction, the use of sparse autoencoder for dimensionality reduction, and subsequent use of the Feed-Forward Neural Network for classification of COVID-19 from non-COVID-19 images. The models are trained and tested using 1046 (504 COVID-19 and 542 non-COVID-19) images obtained from two public accessible datasets. The performance evaluation of the models has placed the combination of InceptionResNetV2 as the best performing model, with 0.9578 accuracy and 0.9821 AUC.

The study conducted by Nayak et al. [100] proposed an automated detection of COVID-19 from X-ray images. The study evaluated eight TL models, which include ResNet34, ResNet-50, MobileNet. InceptionV3, SqueezeNet, AlexNet, VGG16, and GoogleNet for binary classification of COVID-19 and normal cases. The models are trained and tested using datasets obtain from public domains, which include datasets prepared by JP Cohen, Covid-chest-X-ray, and ChestX-ray8 datasets with a total of 703 (500 normal and 203 COVID-19 images). In order to help expand the number of training images, data augmentation techniques were implemented, which include flipping, rotation, scaling, and Gaussian noise. The comparison between model performances has shown that ResNet34 achieved the highest accuracy with 98.33%, a precision of 96.77%, a specificity of 96.67%, and a 0.9836 AUC and 0.9836 F1-score.

Another study that utilized several pretrained models is provided by Narin et al. [101]. The study applied five TL models, which include InceptionV3, Inception-ResNetV2, ResNet50, ResNet101 and ResNet152. The study conducted several binary classifications, which include COVID-19 vs. healthy cases, COVID-19 vs. viral pneumonia, and COVID-19 vs. bacterial pneumonia, using several datasets curated from publicly accessible domains. The comparison between model performances revealed that ResNet50 achieved the best results with 96.1% accuracy on the first dataset, 99.5% on the second dataset, and 99.7% on the third dataset.

The use of CAD of COVID-19 from X-ray images is proposed by Naseer et al. [102]. The study applied two networks, which include the Artificial Neural Network (ANN) and the Artificial Recurrent Neural Network (Long–Short Term Memory (LSTM)) network. In order to maximize the amount of training data, the study conducted several data augmentation processes, which included image enhancement, color transformation, geometric transformation, and noise injection, which yielded 3220 images. Training of the models revolves around three phases, which include training using raw CXR images, training using pre-processed images, and training using enhanced images. The classification process relies on the use of CNN as a feature extractor, which is fed into the LSTM network for classification. The performance evaluation outcome of the joined CNN-LSTM model yielded 99.02% accuracy, 100% sensitivity, and 99% specificity.

(B)Ternary

The binary and ternary classification of X-ray images of COVID-19, non-COVID-19, and viral pneumonia using pretrained models was proposed by Aziz et al. [103]. The detection process follows the use of the connected layer of the ResNetV50V2 model for feature extraction, the use of reduction methods for reduction of feature dimensions, and the use of Gaussian SVM for classifications. The TL models are trained and tested using a dataset acquired from Cohen and Morrison, 2020, with 874 images (254 COVID-19, 310 non-COVID-19, and 310 viral pneumonia). In order to increase the number of training sets, data augmentation was conducted via flipping, rotation, shearing, and height and weight shift. The result of the model performance evaluation yielded 99.5% accuracy for binary classification and 95.5% for ternary classifications.

The ternary classification of X-ray images using a DL model known as CVDNet was proposed by Ouchicha et al. [104]. The model is designed based on a residual neural network, which is constructed using two parallel levels with different filter sizes in order to capture both global and local features of the input datasets. The study trained and validated the model using datasets downloaded from online repositories, which include viral pneumonia (1345), COVID-19 pneumonia (219), and normal cases (1341). The performance evaluation of CVDNet based on 5k-fold cross validation resulted in an average accuracy of 96.69%, 96.84% recall, 96.72% precision, and 96.68% F1-score for three-way classification.

The use of a CNN-based DL fusion framework for the ternary classification of COVID-19 and non-COVID-19 cases was proposed by Shorfuzamman et al. [105]. The study transfer weight (parameters) of three TL models, which include VGG16, ResNet50V2, and GoogleNet (InceptionV3), which are combined into a single model in order to extract images and classify the images using a custom classifier, and the subsequent use of gradient-weighted class activation mapping, in order to view the infected zones. Apart from the use of model performance metric evaluations, the study also conducted cross validation, which resulted in average performances of the model on 1848 image datasets obtained from open domains (Cohen et al. with 616 COVID-19 and Money 2018 with 616 non-COVID-19 viral pneumonia and 616 healthy cases). ResNet50V2 was the best performing model, and achieved an overall accuracy of 95.49%, 99.19% sensitivity, 98.27% specificity, and 95.94% AUC. Workflow of the proposed system is illustrated in Figure 7 (redesigned using Biorender).

(C)Quaternary

The study conducted by Li et al. [106] applied the Cov-Net model for the three-way and four-way classification of COVID-19, non-COVID-19 viral pneumonia, and lung opacities acquired from two public accessible datasets. The first dataset contains three categories, named as D1, while the second dataset contained for classes named as D2. One of the variations of this technique over current techniques is the use of a residual network along with an asymmetric convolution and attention mechanism embedded as the backbone for feature extraction, and the subsequent application of skip-connected dilated convolution with carrying dilation rates in order to attain sufficient feature fusion among low-level detail and high-level semantic information. The performance of the model on the two datasets resulted in 0.9966 and 0.9901 accuracies, respectively.

The study conducted by Ibrahim et al. [18] applied a pretrained AlexNet model for several binary classifications, ternary classifications, and quaternary classifications of X-ray images of COVID-19, viral pneumonia, bacterial pneumonia, and normal cases. The TL model was trained and tested using several datasets curated from online sources. The result of the four-way classification of X-ray images using pretrained AlexNet resulted in an accuracy of 93.42%, sensitivity of 89.18%, and specificity of 98.92%.

Hira et al. [107] applied DL for the binary and multiclass prediction of COVID-19 from X-ray images. The study applied nine DL models, which include AlexNet, Se-ResNet50, ResNeXt-50, Se-ResNeXt-50, ResNet-50, InceptionResNetV2, InceptionV4, GoogleNet, and DenseNet121. The models were trained and validated using several datasets curated from open sources. The performance evaluation of the nine models showed that Se-ResNeXt-50 achieved the best performance for three-way classification, with 97.55% accuracy and 96.89% for four-way classifications.

#### 5.3.2. AI-Powered Detection of COVID-19 from CT Scans

The classification of COVID-19 from non-COVID CT scan images using AI-based CAD was proposed by Syed et al. [89]. Detection of COVID-19 from non-COVID-19 was conducted via four stages, which include curations of CT scans images from two public accessible repositories, which include SARS-COV2-CT (1229 non-COVID-19 and 1252 COVID-19) and community acquired pneumonia (1500 CT images), modification of three pretrained s networks, which include ResNet50, ResNet101 and VGGNet16, a selection of activation function and enhancing firefly algorithms for feature selection, and, finally, the use of a descending order serial approach for fusing optimal selected features and classification using supervised ML, such as SVM classifier. The outcome of the model evaluation has yielded 97.9% accuracy, 97.63% recall, and 97.63% precision, and approximately 34 s of computational time.

The study proposed by Chaddad et al. [108] applied DL-TL for the prediction of COVID-19 from CT scan images. The study applied six DL architectures, which include AlexNet, DarkNet, DenseNet, GoogleNet, NasNet-mobile, and ResNet18. The models are fed with (1) raw datasets and (2) regions of interests corresponding to ground glass opacities, pleural effusion, and consolidation of 100 lung CT images generated from 60 COVID-19 patients. The comparison evaluation of the model performances has shown that DarkNet achieved the best result, with an AUC of 88.16% and accuracy of 82% on a raw dataset, and an AUC of 90.20% and accuracy of 82.30% after incorporating three additional ROIs.

The binary and ternary classification of COVID-19 from CT scan images using two pretrained models was proposed by Mishra et al. [109]. The study applied pretrained VGG16 and ResNet to classify non-COVID-19 pneumonia, COVID-19 pneumonia, and normal cases (400 each in order to achieve class-balanced). The study also conducted data augmentation in order to increase the number of training sets and fine tune the model to optimize its performance. The model is evaluated on the basis of stratified 5k cross validation, and the performance shows that both models achieve more than 99% accuracy for binary classification, while VGG16 achieved 86.74% accuracy and ResNet achieved 88.52% accuracy for ternary classifications.

The study conducted by Katar and Dumman [110] developed a CNN which consists of 19 layers for binary classification of COVID-19 and normal cases from CT scan images. The model was trained (using 1600 images of both positive and negative cases) and tested (using 400 of both positive and negative cases). The performance evaluation of the model resulted in 97.5% accuracy. The study conducted by Kogilavani et al. [111] applied several DL models for binary classification of COVID-19 and normal cases. The study curated 3873 CT (1958 positive cases and 1915 negative cases) scan images, which are partitioned into 70% for training, 15% for testing, and 15% for validation. The images are trained and evaluated using DenseNet-121, EfficientNet, MobileNet, NASNet, Xception, and VGG16. The comparison of the model performances has shown that VGG16 achieved the best result, with 97.68%.

The application of a pretrained model (modified based on random, Bit-S, and Bit-M) for the detection of COVID-19 from over 190 thousand CT scan images collected from 4 thousand patients was proposed by Zhao et al. [112]. The study revolved around the use of pretrained ResNet-V2 (group normalization was replaced with batch normalization and weight standardization for all the convolutional layers) for the classification of COVIDx-CT-2A images into normal and control cases. The evaluation of the model performance resulted in 97.9%, 98.8%, and 99.2% accuracy for Random, Bit-S, and Bit-M, respectively.

The application of a 2D DL approach for the classification of COVID-19 and non-COVID-19 from CT scans images was proposed by Ko et al. [113]. The model, termed as Fast-Track COVID-19 Classification Network (FCONet), was designed using one of the three pretrained networks (Xception inception-V3, ResNet-59, and VGG-16). The designed model was trained using 3993 total images acquired from the Wonkwang University Hospital, Chonnam National University, and Italian Society of Medical and Interventional Radiology public databases. Evaluation of the model performance has shown that FCONet-ResNet-50 achieved the best result, with 99.87% accuracy, 99.58% sensitivity, and 100% specificity.

#### 5.3.3. IoT-Enabled Devices for Detection of COVID-19

Iskanderani et al. [114] proposed an AI/IoT platform for the detection of COVID-19 from Chest X-ray images. The proposed system offers real-time communication and detection of COVID-19 cases. The platform was designed by assembling four DL models, which include DenseNet201, VGG19, InceptionResNetV2, and ResNet152V2. The working principle of the framework revolves around the use of medical sensors to obtain CXR images, which are fed into the ensemble networks for classification. Similarly, Kini et al. [115] proposed the use of an IoT-DL-based framework for the diagnosis of COVID-19 from CT scan images. The system was designed to collect CT scan images using medical IoT devices which transferred the images to an ensemble model (which combined three pretrained networks, which include DenseNet201, InceptionResNetV2, and ResNet152V2). The ensembled model was able to classify CT scan images efficiently on IoT servers.

Le et al. [116] proposed an IoT-enabled depth-wise separable CNN merged with deep SVM for the classification of COVID-19 from X-ray images. The process was dictated by several stages, which included data acquisition using IoT devices which send the images to cloud server, followed by Gaussian filtering to remove noise, feature extraction, and finally classification. Another IoT/DL-enabled framework was proposed by Ahmed et al. [117]. The X-ray images were collected using medical sensors, followed by detection using Faster Region CNN (FR-CNN) and ResNet101 as the backbone network.

Rehman et al. [118] proposed real-time detection of COVID-19 from X-ray images. The framework was developed based on a CNN-residual neural network (ResNet-50). The mechanism behind the real-time CAD revolved around the upload of X-ray images from healthcare centers and remote clinics and subsequent classification using ResNet-50. The performance of the proposed IoT/CAD system achieved 98% accuracy and 0.975 AUC on chest X-ray images acquired from online repositories (already augmented and containing 1824 total images, where 912 are non-COVID-19 and 912 COVID-19 cases).

Punitha et al. [119] proposed a novel e-healthcare platform for diagnosis of COVID-19 using an optimization algorithm. The framework was designed based on the classification approach for the detection of abnormalities in lung CT images via Whale Optimization Algorithms (WOA) optimized Wavelet Neural Network (WNN). The mechanism behind the e-healthcare system revolves around the extraction of the Laws 16 texture energy measures from the preprocessed CT lung images, and subsequent classification using a WNN classifier. Evaluation of the proposed system on publicly accessible datasets resulted in 84.8% accuracy, 82.0% sensitivity, and 73.3% specificity for binary classification of COVID-19 and non-COVID-19 cases.

## 6. Open Research Issue

The increased generation of medical data in healthcare settings has contributed to the development of high-performance models tasked with identifying patterns, extracting features, prediction, and classification of medical data [75,77]. Analyzing the large amount of data generated in medical settings requires the use of reliable, fast, and accurate systems which can relieve the workload of the medical data analyst. The application ML models over the last decade have been shown to address this issue. Several ML models are used in healthcare settings as a form of CAD to assist the physician in conducting accurate diagnoses and in appropriate decision making [71,76].

Despite the wide application of ML models in healthcare settings, they are hindered by several challenges, which include the lack of a sufficient amount of data. Training of ML models using a substantial amount of data is crucial for high performance. In order to address the shortage of data, scientists developed TL, where weights and features are extracted from trained models and repurposed on new tasks with insufficient datasets. The use of TL models, also known as pretrained models, has shown to outperform models developed from scratch [120,121].

Another challenge facing the application of ML models is underfitting and overfitting. Underfitting occurs when ML models perform poorly on both training and testing sets (i.e., when models neither perform well on training dataset nor generalize the new or unseen dataset). Underfitting is associated with high bias and low variance. Overfitting occurs when the ML model performs well on training datasets, but performs poorly on test sets [70,122]. Overfitting is associated with high variance and low bias. The performance of ML models depends on the types of images use [70]. Poor performance can also be related to training images with small amounts of data, or data that contains noise. The use of data augmentation techniques such as rotation, flipping, mirroring, zooming, cropping, etc. increases the number of training sets [123]. Other pre-processing steps are used to remove noise from images, resize images to fit into the models, and extract features that can be classified using classifiers [123,124].

The landscape of AI-powered models is changing from the application of single models to a hybrid of ensembled models. Ensembled models combined predictions from two or more models. Ensembled learning includes parallel ensembled and sequential ensemble methods. Ensembled learning techniques can also be classified as bagging (such as bootstrap aggregation), boosting (gradient boosting machine or GMB, LightGBM, Adaboost, etc.), stacking, and blending. The use of ensembled models is associated with improving performance (originally developed to reduce variance, thereby improving performance) and robustness [4,125,126].

The integration of IoT in the medical field, known as IoMT, is transforming healthcare systems into an interconnected unit which allows wireless exchange of medical data between devices, medical experts, and medical records storage or cloud systems [127]. The integration of IoT with CAD detection, also known as AI/IoT-powered systems, has led to the development of several platforms that enable users to upload medical images (such as CT scans, ultrasound, X-rays, and microscopic slide images) and non-medical images (such as skin and facial images) for real-time diagnosis [83,84].

Despite the prospect of AI/IoT-powered detection, the system is challenged by several factors, which include the cost of deployment, data ownership, privacy, ethical, and security issues. The major concern of employing these devices include device hijacking, cyber-attacks, data theft, etc. In order to address these issues, several medical companies have developed encrypted methods to prevent fraudulent attacks and breaches of privacy. However, despite their efforts, the system is still prone to cyber-attacks and device hijacking. This raises the need for developing a more secured system through encryption, authentication, tracking, and monitoring protocols [82,84].

## 7. Conclusions

The global pandemic witnessed as a result of spread of the COVID-19 disease associated with SARS-CoV-2 has changed the landscape of diseas diagnosis. Several techniques have been developed and repurposed in order to provide accurate and reliable detection of the virus. Molecular testing based on RT-PCR is regarded as the standard approach for the detection of COVID-19, followed by serological antigen-based detection. Despite the high reliance on these approaches, they are limited by so many challenges, which include false positive results, low accuracy, the need of trained pathologists, the need for chemicals, longer processing time, high cost, etc. These factors limit the use of molecular testing in remote areas and underdeveloped countries. This call for the need to provide an alternative approach that can provide accurate results, eliminate the need of toxic chemicals, and reduce the high cost of assays. Thus, healthcare experts turn to medical imaging such as X-rays, CT scans, and lung ultrasounds as alternative or confirmatory testing. However, this approach is also clouded by several challenges, which include tediousness in the case of the interpretation of a large number of cases and misinterpretation.

To address these issues, scientists incorporated CAD using ML models and classifiers. These models have shown to achieve high performance compared to human interpretation. In order to allow real-time testing, scientists developed IoT/AI-enabled systems, or e-healthcare systems, which allow the uploading of medical images and the subsequent classification of cases into binary (COVID-19 and non-COVID-19), three-way, and four-way classification (COVID-19, non-COVID-19 viral pneumonia, bacterial pneumonia, and healthy cases). Thus, this review provides extensive knowledge of the state-of-the-art detection of COVID-19 using molecular testing, CAD, and IoT/AI-powered detection.

## Figures and Tables

**Figure 1 sensors-23-00426-f001:**
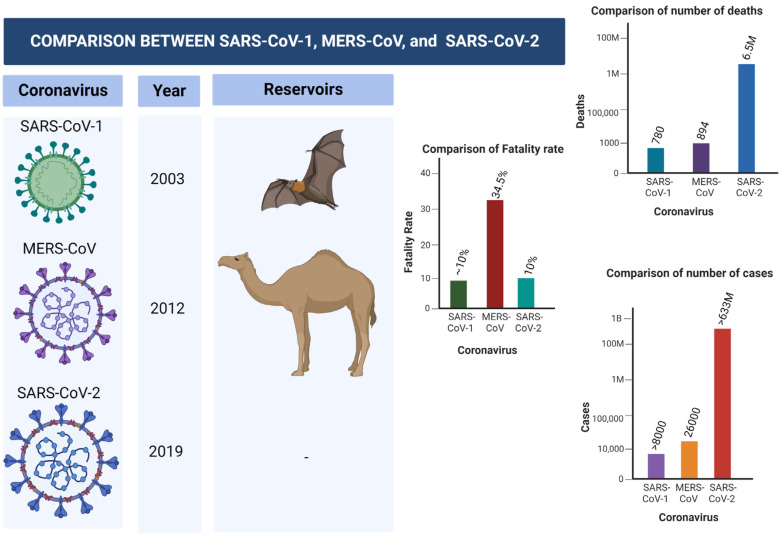
Comparison between SARS-CoV-1, MERS-CoV-2, and SARS-CoV-2.

**Figure 2 sensors-23-00426-f002:**
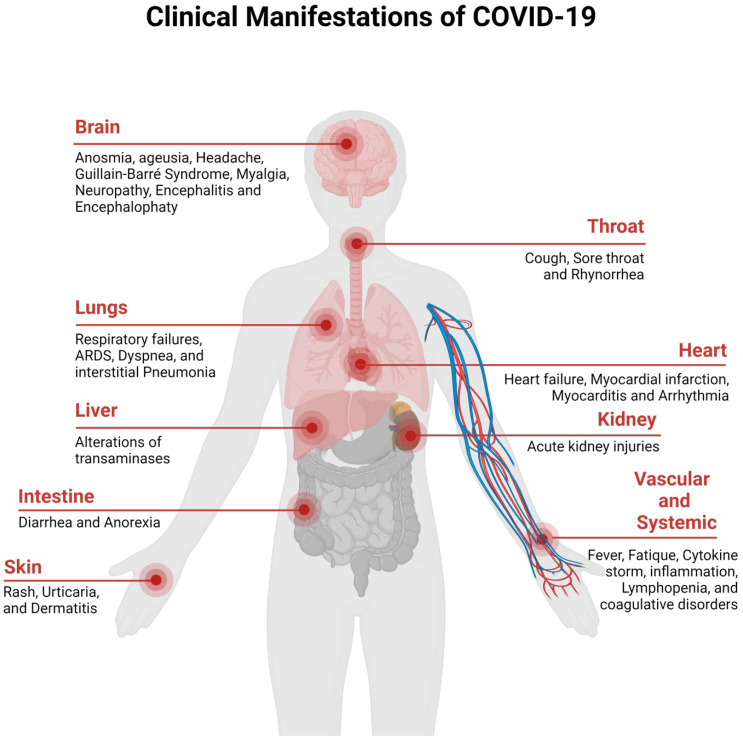
Clinical Manifestation of COVID-19.

**Figure 3 sensors-23-00426-f003:**
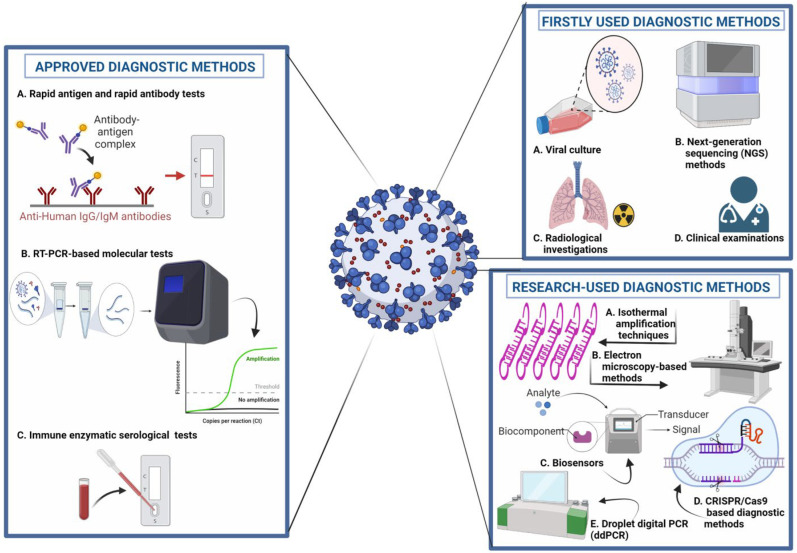
Molecular diagnostic approaches for the detection of COVID-19. 1. Approved Diagnostic Methods; A. Rapid Antigen and Rapid Antibody Tests; B. RT-PCR-based Molecular Tests; C. Immune Enzymatic Serological Tests. 2. Firstly Used Diagnostic; A. Viral Culture; B. Next Generation Sequencing (NGS) Methods; C. Radiological Investigation; D. Clinical Examination; 3. Research-used Diagnostic Methods; A. Isothermal Amplification Techniques; B. Electron microscopy-based Methods; C. Biosensors; D. CRISPR/Cas9-based Diagnostic Methods. E. Droplet digital PCR (ddPCR).

**Figure 4 sensors-23-00426-f004:**
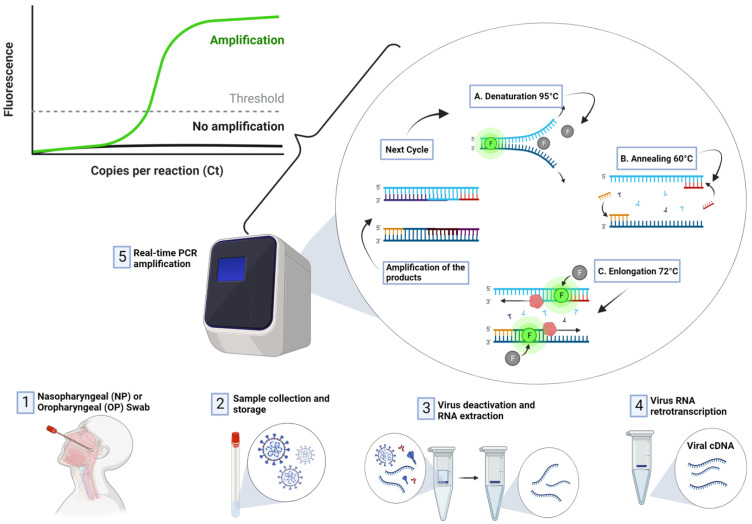
Detection of COVID-19 using RT-PCR Technique.

**Figure 5 sensors-23-00426-f005:**
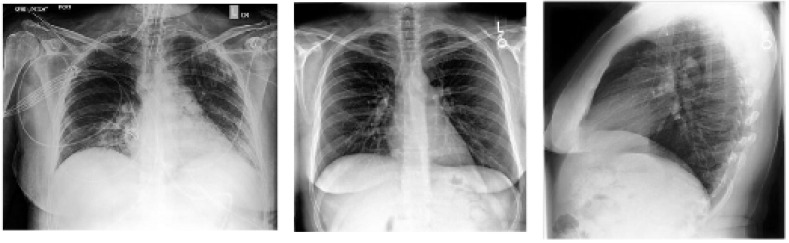
**Left**: Anterior–posterior (AP) view chest X-ray. **Middle**. Posterior–anterior (PA) view frontal chest X-ray. **Right**: lateral chest X-ray.

**Figure 6 sensors-23-00426-f006:**
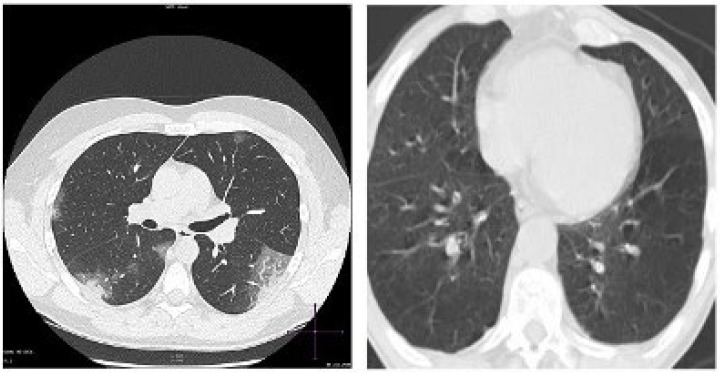
CT scan Image. **Left**: COVID-19. **Right**: Normal.

**Figure 7 sensors-23-00426-f007:**
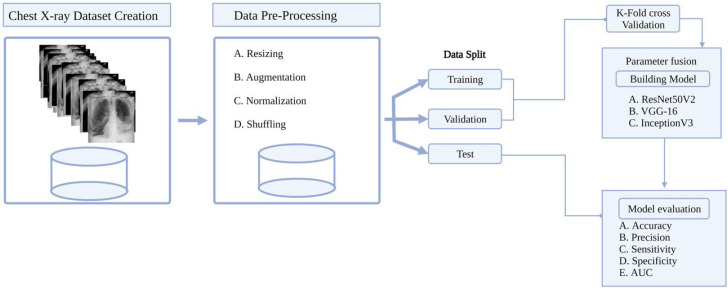
AI-Powered Detection of COVID-19 from X-ray images.

**Table 1 sensors-23-00426-t001:** Comparison with similar studies.

Reference	COVID-19 Pandemic	Molecular Diagnostic and Biosensors	Medical Imaging	AI, ML, DL and TL	IoT/IoMT	Comparison	Open Research Issue
[25]	✓	✓	-	-	-	-	✓
[26]	✓	✓	-	-	-	-	-
[27]	✓	✓	-	-	-	-	✓
[28]	✓	-	✓	✓	-	-	✓
This review	✓	✓	✓	✓		✓	✓

## Data Availability

Not applicable.

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
