# Peer review of "Current Technologies for Detection of COVID-19: Biosensors, Artificial Intelligence and Internet of Medical Things (IoMT): Review"

_sensors, 2022, doi:10.3390/s23010426_

Round 1

Reviewer 1 Report

The authors summarize research in the field of medicine using the application of AI and IoMT methods to identify the disease COVID 19.

The topic is very current, but the publication has several shortcomings.
I recommend improving the following parts of the publication:
1. The article lacks an analysis of the presented methods and a better explanation of when to use which method and pointing out its strengths and weaknesses.
2. It would be appropriate to indicate the data sources to which the methods were applied.
3. In the article, it would help to expand the analytical part with the methodologies used by individual countries in data collection, which theoretically can affect the quality of the assessment.
4. In chapter 2 ( COVID-19), it would be better to show using a graph instead of table 2.
5. The article lacks a technical part. A chapter on IoT deployment research is missing. What advantages does IoT provide? What disadvantages do IoT provide? What security risks result from the deployment of IoT devices? Since we are working with sensitive patient data.
6. I suggest increasing the number of studies and adding a new discussion of the security risks of IoT deployment to show the benefit.
7. Perhaps consider the following studies.
8. DDoS reflection attack based on IoT: A case study
9. Mitigation against DDoS Attacks on an IoT-Based Production Line Using Machine Learning
10. The Vulnerability of the Production Line Using Industrial IoT Systems under DDoS Attack
12. Some sentences are too long to follow, so it is recommended to break them into short but meaningful ones to make the manuscript readable.
13. Quite a lot of abbreviations are used in the manuscript. To improve readability, it is recommended to use a table to locate all frequently used abbreviations with their descriptions.
14. What are the disadvantages of your research?

Author Response

Reviewer 1

Reviewer’s Comments: The article lacks an analysis of the presented methods and a better explanation of when to use which method and pointing out its strengths and weaknesses.

Response: The strengths of molecular techniques are provided. However, we already explained that Medical imaging and CAD system are use as confirmatory approach while Molecular testing is the most preferred approach.

3.2 Strengths and Weakness of Molecular Testing

Currently there are several techniques developed for the detection of virus. Among these techniques, molecular testing is the most prepared approach due to it high sensitivity and specificity. These molecular test include, antibody, antigen and RT-PCR which can also be subdivided into molecular test and serological or antibody test. Molecular testing revolves around the detection of viral RNA in human body while the virus is still replicating. While antibody assays detect the presence of antibodies produced as a result of human immune response against the virus…’

Reviewer’s Comments: It would be appropriate to indicate the data sources to which the methods were applied.

Response: Collection of several or most popular datasets are discussed in sub-section 5.1.1.

5.1.1 Radiographic Dataset

Ever since the WHO organization declared COVID-19 as global pandemic, medical experts have curated several radiographic datasets from clinical settings into an online repository. Kaggle and GitHub are among 2 of the most popular domains that are easily accessible. These repositories contain thousands of radiographic images of both X-ray and CT scan images of bacterial pneumonia, COVID-19 pneumonia, non-COVID-19 viral pneumonia and healthy cases …’

Reviewer’s Comments: In the article, it would help to expand the analytical part with the methodologies used by individual countries in data collection, which theoretically can affect the quality of the assessment.

Response: This issue is highlighted under the Radiographic datasets.

One of the challenges of using collection of more than one dataset is the likelihood of reputation, diversity of images acquired from different types of devices and settings…’

Reviewer’s Comments: In chapter 2 (COVID-19), it would be better to show using a graph instead of table 2.

Response: We replaced the table with a figure.

Reviewer’s Comments: The article lacks a technical part. A chapter on IoT deployment research is missing. What advantages does IoT provide? What disadvantages do IoT provide? What security risks result from the deployment of IoT devices? Since we are working with sensitive patient data. I suggest increasing the number of studies and adding a new discussion of the security risks of IoT deployment to show the benefit. Perhaps consider the following studies.

DDoS reflection attack based on IoT: A case study

Mitigation against DDoS Attacks on an IoT-Based Production Line Using Machine Learning

The Vulnerability of the Production Line Using Industrial IoT Systems under DDoS Attack

Response: Deployment, advantage and disadvantages of IoT-based systems and the suggested references are added.

4.3.1 Advantage of IoT-based Systems: How IoT is shaping Clinical diagnosis

Recent advances in sensing technology, IT and software engineering and its adoption in healthcare settings have made remote monitoring, real-time diagnosis, analysis and sharing of data possible. This technology has shown to contribute in making diagnosis more efficient and safer as well as aiding medical experts in making appropriate diagnosis. IoT/AI-driven models are applied in medical settings at massive scales in order to relieve the intensive workload of medical experts, increase performance, efficiency and minimize long diagnosis processes…’

4.3.2 Disadvantage of IoT-based Systems:

Despite the fact that IoT offer several benefits in healthcare settings, however one of the major challenges limiting its application is security threat such as data theft, device hijacking, system attacks (e.g., Distributed Denial of Service or DDoS), data ownership disputes etc. The interconnection between IoT-based devices with internet make them prone to cyber-attacks or hacks…’

4.3.3 Deployment of IoT-Based Systems

Deployment of IoT system for diagnosis of pathological diseases require several features which include medical data (such as images acquired from CT scans, X-rays, MRIs, PETs, SPECTs, hybrid systems, electrophysiological devices, human faces, skin, etc.) AI-driven models train and validated using large amount of dataset and website that can be used to deploy the model for real-time classifications…’

Reviewer’s Comments: Some sentences are too long to follow, so it is recommended to break them into short but meaningful ones to make the manuscript readable.

Response: Some of the sentences are shorten as recommended.

Reviewer’s Comments: Quite a lot of abbreviations are used in the manuscript. To improve readability, it is recommended to use a table to locate all frequently used abbreviations with their descriptions.

Response: Abbreviation list are added as recommended.

Reviewer’s Comments: What are the disadvantages of your research?

Response: Since this research is a review article that is aim to provide readers with state-of-the-art knowledge of COVID-19 diagnosis approaches, we can’t think of any disadvantage.

Reviewer 2 Report

I have attached my comments.

Author Response

Reviewer 2

This paper provides a holistic approach to COVID-19 detection based on (1) molecular diagnosis, which includes RT-PCR, antigen-antibody and CRISPR-based biosensors and (2) computer-aided detection based on AI-driven models, which include Deep Learning and Transfer learning approach. What are the contributions of this work in the field of COVID-19 detection? Almost all information is about related works. This paper is similar to a review report without experimental results. I have the following concerns:

Reviewer’s Comments: Format of table 1 needs to be corrected; adjust it according to the journal's templates. Furthermore, add the scientist's name in the "Ref" column.

Response: The names of the scientists are added in the ref column.

Reviewer’s Comments: The logic of Sections 1 and 2, as well as the research direction, could be more precise.

Response: The manuscript is a review article, that is why we provide thorough explanation

Reviewer’s Comments: What is the main contribution and novelty of this work? Please list them in the Introduction section.

Response: The main aim or contribution of this work is provided under 1.2 Scope.

The main aim of this review is to provide holistic approach on emerging technologies that aid in the detection of COVID-19 such as RT-PCR, antigen-antibody and CRISPR-based biosensor as well as computer aided detection using AI-driven models. Moreover, the review also covers the integration of IoMT and AI for the development of smart system for the detection of the disease.

Reviewer’s Comments: Please change the title of Section 2 to "related works" or "literature reviews". Authors have to rearrange Sections 1 and 2 to make them explicit. For example, the definition of COVID-19 can be described in Section 1.

Response: The reason why we named chapter 2 or section as COVID-19 instead of Related work is because it is a review article.

Reviewer’s Comments: Is figure 1 the author's work? If not, please cite it according to copyright issues.

Response: We design the figures using Biorender

Reviewer’s Comments: Please add a figure to describe the proposed method flow chart in Section 3.

Response: There is no proposed methods since we did not conduct any experiments

Reviewer’s Comments: It is necessary to give some details of the proposed model by adding figures in Section 3.

Response: There is no proposed methods since we did not conduct any experiments

Reviewer’s Comments: In Section 4., add more explanation of the proposed model to enrich this work and contributes to the field.

Response: There is no proposed methods since we did not conduct any experiments

Reviewer’s Comments: There needs to be an experimental section and results.

Response: There is no proposed methods since we did not conduct any experiments

Reviewer’s Comments: Please add information about the experimental environment, for example, the specification of computer and software, training process, and size of images.

Response: There is no proposed methods since we did not conduct any experiments

Reviewer’s Comments: Comparing this work with other similar works is necessary. Thus, please add tables or figures to prove this work's advantages over other similar works.

Response: There is no proposed methods since we did not conduct any experiments

Reviewer’s Comments: In general, Section 5 must be modified with more evaluations and comparisons with other popular methods.

Response: There is no proposed methods since we did not conduct any experiments

Reviewer’s Comments: Please add a "Limitation and Discussion" section to discuss the proposed method's limitations and future research gaps in this field.

Response: There is no proposed methods since we did not conduct any experiments

Reviewer’s Comments: Authors should also check the article for typo errors and English grammar.

Response: The whole manuscript is proofread again.

Reviewer’s Comments: Please check the style and format of the references.

Response: We used the authors guidelines to prepare the references

Reviewer 3 Report

Despite the fact that COVID-19 is no longer a global pandemic due to the development and integration of different technologies for the diagnosis and treatment of the disease. Technological advancement in the field of molecular biology, electronics, computer science, artificial intelligence, the Internet of Things, nanotechnology, etc., has led to the development of molecular approaches and computer-aided diagnosis for the detection of COVID-19. This study provides a holistic approach on COVID-19 detection based on (1) molecular diagnosis, which includes RT-PCR, antigen-antibody, and CRISPR-based biosensors, and (2) computer-aided detection based on AI-driven models, which include Deep Learning and Transfer learning approach. The review also provides a comparison between these 2 emerging technologies and open research issues for the development of smart-IoMT- enable a platform for the detection of COVID-19.  Although the data presented in this review is enough but still needs the following minor revision:

- " In order to control the disease, scientists from different field work hand in hand together to develop diagnosis approaches, prediction models, treatment control strategies, vaccines, etc."

- add the following two works for the support of this statement: i) COVID-19 Classification using Chest X-Ray Images based on Fusion Assisted Deep Bayesian Optimization and Grad-CAM Visualization, ii) A Healthcare System for COVID19 Classification Using Multi-Type Classical Features Selection

- "Medical experts rely on 2 main molecular approaches, which include RT-PCR and antibody-antigen based techniques for the detection of the disease. However, among these two molecular testing approaches, RT-PCR is regarded as the gold standard technique due to it specificity and accuracy."- add the reference for this statement. 

- "Despite the higher specificity of these molecular approaches, they are hindered by several challenges, including false positive results, which can lead to miss-diagnosis and expensive, especially in underdeveloped countries. As an alternative, healthcare professionals employ radiographic screening using X-ray imaging and CT-scan imaging, which allow scientists to discriminate between positive and negative cases."- add the following two works for this statement: i) COVID19 Classification using Chest X-Ray Images: A Framework of CNN-LSTM and Improved Max Value Moth Flame Optimization

- Enhance the related work by adding the following articles: i) COVID-19 Classification from Chest X-Ray Images: A Framework of Deep Explainable Artificial Intelligence, ii) A Rapid Artificial Intelligence-based Computer-Aided Diagnosis System for COVID19 Classification from CT Images, iii) Residual Attention Deep SVDD for COVID-19 Diagnosis Using CT Scans

- It is better to discuss the COVID-19 datasets under a separate heading and then discuss their results and gaps. 

- Add reference for Fig. 1. Similarly for Fig 2 and 3. 

- The open research issues need to be further enhanced by reading more recent articles like: i) COVID-19 Case Recognition from Chest CT Images by Deep Learning, Entropy-Controlled Firefly Optimization, and Parallel Feature Fusion, ii) Deep Rank-Based Average Pooling Network for Covid-19 Recognition

Author Response

Reviewer 3

Reviewer’s Comments: " In order to control the disease, scientists from different field work hand in hand together to develop diagnosis approaches, prediction models, treatment control strategies, vaccines, etc." add the following two works for the support of this statement: i) COVID-19 Classification using Chest X-Ray Images based on Fusion Assisted Deep Bayesian Optimization and Grad-CAM Visualization, ii) A Healthcare System for COVID19 Classification Using Multi-Type Classical Features Selection

Response: The references suggested are cited.

Reviewer’s Comments: "Medical experts rely on 2 main molecular approaches, which include RT-PCR and antibody-antigen based techniques for the detection of the disease. However, among these two molecular testing approaches, RT-PCR is regarded as the gold standard technique due to it specificity and accuracy."- add the reference for this statement.

Response: The reference is added as suggested.

Reviewer’s Comments: "Despite the higher specificity of these molecular approaches, they are hindered by several challenges, including false positive results, which can lead to miss-diagnosis and expensive, especially in underdeveloped countries. As an alternative, healthcare professionals employ radiographic screening using X-ray imaging and CT-scan imaging, which allow scientists to discriminate between positive and negative cases."- add the following two works for this statement: i) COVID19 Classification using Chest X-Ray Images: A Framework of CNN-LSTM and Improved Max Value Moth Flame Optimization

Response: The references are added as suggested.

Reviewer’s Comments: Enhance the related work by adding the following articles: i) COVID-19 Classification from Chest X-Ray Images: A Framework of Deep Explainable Artificial Intelligence, ii) A Rapid Artificial Intelligence-based Computer-Aided Diagnosis System for COVID19 Classification from CT Images, iii) Residual Attention Deep SVDD for COVID-19 Diagnosis Using CT Scans

Response: The references are added as suggested except the (iii) one which we couldn’t find on Google scholar.

Reviewer’s Comments: It is better to discuss the COVID-19 datasets under a separate heading and then discuss their results and gaps.

Response: We provided 7 COVID-19 datasets that are mostly used for CAD of COVID-19 and non-COVID-19 pneumonia.

5.1.1 Radiographic Dataset

Ever since the WHO organization declared COVID-19 as global pandemic, medical ex-perts have curated several radiographic datasets from clinical settings into an online repository. Kaggle and GitHub are among 2 of the most popular domains that are easily accessible. These repositories contain thousands of radiographic images of both X-ray and CT scan images of bacterial pneumonia, COVID-19 pneumonia, non-COVID-19 viral pneumonia and healthy cases. One of the challenges of using collection of more than one dataset is the likelihood of reputation, diversity of images acquired from different types of devices and settings…’

Reviewer’s Comments: Add reference for Fig. 1. Similarly, for Fig 2 and 3.

Response: We designed the figures using Biorender

Reviewer’s Comments: The open research issues need to be further enhanced by reading more recent articles like: i) COVID-19 Case Recognition from Chest CT Images by Deep Learning, Entropy-Controlled Firefly Optimization, and Parallel Feature Fusion, ii) Deep Rank-Based Average Pooling Network for Covid-19 Recognition

Response: The references are cited under open research issue

Reviewer 4 Report

1-     In this area, a lot of work has been done, and a lot of important work is missing.

2-     Introduction section needs more investigation of some latest and relevant work,

3-     Table 2 should be revised. 

4-     It will be better to understand if you Explain all abbreviations separately (Table form).

5-     There are ambiguities and unclear meanings throughout the whole paper. Be more precise with language.

6-     In page 2 rewrte this sentence “Despite the higher specificity of these molecular approaches, they are hindered by 48 several challenges which include false positive results which can lead to miss-diagnosis 49 and expensive especially in underdeveloped countries.”

7-     Chapter 2 discusses about COVID-19 pandemic. Chapter 3 present an overview on 114 molecular approaches for the detection of COVID-19…. You should write section 2, section 3(instead of chapter2 etc,) at the end of the introduction part.

8-     In page 8-line 252 Song et al., [46] i think it should be Song et al. [46]

Author Response

Reviewer 4

Reviewer’s Comments: In this area, a lot of work has been done, and a lot of important work is missing.

Response: We added more studies as suggested.

Reviewer’s Comments: Introduction section needs more investigation of some latest and relevant work.

Response: The Introduction section is designed to introduced the topic, scope and similar reviews. It has limitation, that is why we provide more relevant work in the subsequent sections

Reviewer’s Comments: Table 2 should be revised.

Response: We replaced the table with a figure.

Reviewer’s Comments: It will be better to understand if you Explain all abbreviations separately (Table form).

Response: The list of abbreviations is added as suggested

Reviewer’s Comments: There are ambiguities and unclear meanings throughout the whole paper. Be more precise with language.

Response: We tried as much as possible to correct this issue.

Reviewer’s Comments: In page 2 rewrte this sentence “Despite the higher specificity of these molecular approaches, they are hindered by several challenges which include false positive results which can lead to miss-diagnosis and expensive especially in underdeveloped countries.”

Response: The paragraph is adjusted as suggested.

Even though molecular techniques such as RT-PCR and antibody testing are re-garded as the standard procedures for the detection of SARS-CoV-2, they are hindered by several challenges which include incidence of false positive results which can lead to miss-diagnosis. These testing procedures are also costly especially in remote areas and countries with substandard healthcare system.

Reviewer’s Comments: Chapter 2 discusses about COVID-19 pandemic. Chapter 3 present an overview on 114 molecular approaches for the detection of COVID-19…. You should write section 2, section 3(instead of chapter2 etc,) at the end of the introduction part.

Response: The “Chapters” are changed to “Sections” as suggested.

Reviewer’s Comments: In page 8-line 252 Song et al., [46] i think it should be Song et al. [46]

Response: The In-text citation is adjusted as suggested.

Round 2

Reviewer 1 Report

All my concerns have been addressed. The paper is well revised and can be accepted.

Author Response

No concern raised by the reviewer

Reviewer 2 Report

The authors answered all comments and revised the manuscript.

Author Response

No further concern raised by the reviewer

Reviewer 4 Report

The author has improved greatly, but I still have more concerns to solve before publication.

1-   Table 2 need to be formatted carefully.

2-   Section 3.1 and 3.1.1. You should add something in section 3.1 or remove it.

3-        Reference should be in a unified format and carefully rechecked, and can add the most cited literature; “Deep learning model integrating features and novel classifiers fusion for brain tumor segmentation”

4-      Open research issue part you should more information.

Author Response

Reviewer’s Comment: Table 2 need to be formatted carefully.

Response: The table is formatted as suggested

Reference

COVID-19 pandemic

Molecular Diagnostic and Biosensors

Medical Imaging

AI, ML, DL and TL

IoT/IoMT

Comparison

Open Research Issue

 [22]

✓

✓

-

-

-

-

✓

 [23]

✓

✓

-

-

-

-

-

 [24]

✓

✓

-

-

-

-

✓

 [25]

✓

-

✓

✓

-

-

✓

Our Report

✓

✓

✓

✓

✓

✓

Reviewer’s Comment: Section 3.1 and 3.1.1. You should add something in section 3.1 or remove it.

Response: Section 3.1 and sub-section 3.1.1 are very crucial to this article. However, we add a paragraph to elaborate on laboratory testing. Part of section include subsequent sub-sections which we explain the most common laboratory-based procedures which include RT-PCR, antigen and antibody techniques.

3.1 Laboratory Assays

Accurate, sensitive and rapid laboratory assays for the detection of SARS-CoV-2 is crucial for treatment and control of COVID-19 infection. Currently, there are myriad of tests available in the market. However, the adoption of appropriate laboratory testing techniques and type of specimen (nasopharyngeal aspirates, nasopharyngeal swabs, mid-turbinate swabs, oropharyngeal swab etc.) is one of the cornerstones for the timely management and control of the disease. The literature encompasses several studies focusing on laboratory testing procedures which include nucleic acid amplifications, antigen test, antibody tests and point of care testing. These procedures are currently employed in detection of SARS-CoV-2 in clinical diagnosis of symptomatic patients, asymptomatic population screening, contact investigations, targeted high-risk population screening, retrospective population screening, monitoring of infectivity, disease severity monitoring etc. [35, 36, 127].

Reviewer’s Comment: Reference should be in a unified format and carefully rechecked, and can add the most cited literature; “Deep learning model integrating features and novel classifiers fusion for brain tumor segmentation”

Response: References are unified and the suggested article is sighted as [126]. However, some these references lack page numbers, volume numbers etc.

Reviewer’s Comment: Open research issue part you should more information.

Response: We added two paragraphs on AI/IoT-powered systems, their advantage and limitations which include security issues.

The integration of IoT in medical field known as IoMT is transforming healthcare systems into an interconnected unit which allow wireless exchange of medical data between devices, medical experts and medical records storage or cloud systems [128]. The integration of IoT with CAD detection also known as AI/IoT-powered system has led to development of several platforms that enable users to upload medical images (such as CT scan, ultrasound, X-ray, microscopic slide images) and non-medical images (such as skin and facial images) for real-time diagnosis [110-111].

Despite the prospect of AI/IoT-powered detection, the system is challenged by several factors which include the cost of deployment, data ownership, privacy, ethical and security issues. The major concern of employing these devices include device hijacking, cyber-attacks, data theft etc. In order to addressed these issues, several medical companies developed encrypted methods to prevent fraudulent attacks and breach of privacy. However, despite their efforts, the system is still prone to cyber-attacks and device hi-jacking. This raises the need for developing a more secured system through encryption, authentication, tracking and monitoring protocols [109, 111].
